# ISLS: An Illumination-Aware Sauce-Packet Leakage Segmentation Method

**DOI:** 10.3390/s24103216

**Published:** 2024-05-18

**Authors:** Shuai You, Shijun Lin, Yujian Feng, Jianhua Fan, Zhenzheng Yan, Shangdong Liu, Yimu Ji

**Affiliations:** 1School of Internet of Things, Nanjing University of Posts and Telecommunications, Nanjing 210023, China; xiao_yihong@163.com (S.Y.); fengyujian_904@163.com (Y.F.); 2School of Computer Science, Nanjing University of Posts and Telecommunications, Nanjing 210023, China; linshijun2767@foxmail.com (S.L.); lsd@njupt.edu.cn (S.L.); 3The 63rd Research Institute, National University of Defense Technology, Nanjing 210007, China; fjh7659@163.com; 4Northern Information Control Research Academy Group Co., Ltd., Nanjing 211153, China; yanzhenzheng_1314@163.com

**Keywords:** sauce-packet leakage segmentation, uneven illumination, multi-level feature fusion, attention mechanism

## Abstract

The segmentation of abnormal regions is vital in smart manufacturing. The blurring sauce-packet leakage segmentation task (BSLST) is designed to distinguish the sauce packet and the leakage’s foreground and background at the pixel level. However, the existing segmentation system for detecting sauce-packet leakage on intelligent sensors encounters an issue of imaging blurring caused by uneven illumination. This issue adversely affects segmentation performance, thereby hindering the measurements of leakage area and impeding the automated sauce-packet production. To alleviate this issue, we propose the two-stage illumination-aware sauce-packet leakage segmentation (ISLS) method for intelligent sensors. The ISLS comprises two main stages: illumination-aware region enhancement and leakage region segmentation. In the first stage, YOLO-Fastestv2 is employed to capture the Region of Interest (ROI), which reduces redundancy computations. Additionally, we propose image enhancement to relieve the impact of uneven illumination, enhancing the texture details of the ROI. In the second stage, we propose a novel feature extraction network. Specifically, we propose the multi-scale feature fusion module (MFFM) and the Sequential Self-Attention Mechanism (SSAM) to capture discriminative representations of leakage. The multi-level features are fused by the MFFM with a small number of parameters, which capture leakage semantics at different scales. The SSAM realizes the enhancement of valid features and the suppression of invalid features by the adaptive weighting of spatial and channel dimensions. Furthermore, we generate a self-built dataset of sauce packets, including 606 images with various leakage areas. Comprehensive experiments demonstrate that our ISLS method shows better results than several state-of-the-art methods, with additional performance analyses deployed on intelligent sensors to affirm the effectiveness of our proposed method.

## 1. Introduction

With the advancement of computer vision technology, abnormal region segmentation has become crucial for intelligent industrial production [1,2,3]. Sauce-packet leakage segmentation is a typical abnormal region segmentation task. Sauce-packet leakage is typically due to improper machine settings on the production line. For example, inappropriate temperature and time settings of the high-temperature sealing machine or mismatched speed between the rotating shaft and the sauce-filling machine nozzles can lead to leakage at the connection of the sauce packets. However, in actual sauce-packet production lines, the image quality is affected by blurring due to uneven illumination (i.e., overexposure or insufficient illumination), which affects segmentation performance. The blurring sauce-packet leakage segmentation task (BSLST) can alleviate uneven illumination, which aims to identify the pixel-level leakage at the sauce-packet connection [4]. Traditional and most deep learning algorithms have insufficient performance on the BSLST, due to challenges in handling pixel-level classification under uneven illumination conditions, and those algorithms lack sufficient granularity in capturing monotonous features [5,6,7]. The solving of the BSLST can enhance the efficiency of sauce production in the catering industry, which would reduce the miss rate of manual inspections and unleash economic vitality. Hence, to facilitate the industrialization process, researching the BSLST is particularly important.

The leakage segmentation of sauce packets can be divided into traditional methods and Convolutional Neural Network (CNN) methods [8]. Traditional methods rely on expert experience. Songming et al. [9] developed an improved detector using the Canny operator, which improves the computational efficiency and increases the precision of fiber identification. Sharma et al. [10] used a Histogram of Oriented Gradient (HOG) and Support Vector Machine (SVM) segmentation method, which integrates with a modified ResNet50 model for brain tumor detection to help clinicians. Similarly, Hongbin et al. [11] proposed a segmentation method by HOG and Local Binary Pattern (LBP), which combines both HOG and LBP features to accurately identify crack anomalies. Binwu et al. [12] developed a secondary template-matching method, which extracted the Region of Interest (ROI) by using the four-threshold algorithm. However, traditional methods rely on hand-crafted features, and the generalization performance in real scenes is insufficient.

The CNN methods automatically extract discriminative features and do not rely on expert experience. Wang et al. [13] proposed HRNet, which connects the high-to-low-resolution convolution streams in parallel and repeatedly exchanges the information across resolutions. Yu et al. [14] proposed BiseNetv2, which involves a detail branch and a semantic branch. Xie et al. [15] proposed SegFormer, which unifies transformers with lightweight multilayer perception decoders. Cao et al. [16] proposed Swin-Unet, which designs a novel pure transformer-based U-shaped encoder–decoder for medical image segmentation. Zhou et al. [17] presented a water leakage detection method under insufficient illumination conditions and uneven illumination on the turbine layer, which effectively detects water dripping and the leakage of turbine floor equipment by a wheel inspection machine. Huang et al. [18] introduced tunnel lining surface crack and leakage defect detection under uneven illumination conditions. Shen et al. [19] proposed a cofe-Net to suppress global degradation factors under uneven illumination, image blurring, and artifacts, while simultaneously preserving anatomical retinal structures and pathological characteristics. Zhao et al. [20] proposed a shield tunnel lining leakage segmentation approach to identify leakage areas and scaling images. Hai et al. [21] developed an R2RNet to enhance images under low illumination, including image decomposition, denoising, contrast amplification, and the meticulous preservation of intricate details. However, the above methods have not been applied to the field of sauce-packet leakage segmentation.

To solve those problems, we propose the two-stage illumination-aware sauce-packet leakage segmentation (ISLS) method. To the best of our knowledge, we found that our ISLS method is the first to detect sauce-packet leakage under uneven illumination. Firstly, in the illumination-aware region enhancement stage, an efficient localization algorithm [22] is introduced to reduce the calculation of invalid areas. And we designed the uneven-light image-enhancement (ULIE) method to alleviate the problems of blurred images under uneven illumination conditions. Specifically, the ULIE method is built upon the retinex model to enhance image clarity under insufficient illumination condition. And the ULIE method utilizes contrast-limited adaptive histogram equalization to alleviate the leakage details in overexposure images. Secondly, in the leakage region segmentation stage, to effectively relieve the problem of missing information, we propose the multi-scale feature fusion module (MFFM) for capturing multi-scale discriminative representation. Our MFFM fuses a variety of feature maps from top to bottom. And the resulting fused feature map serves as the input to our proposed network decoder, thereby enhancing the decoder’s semantic recovery performance. Finally, the Sequential Self-Attention Mechanism (SSAM) utilizes a sequential structure, which combines the channel and spatial attention mechanisms, thereby achieving the effective mining of salience information. In summary, our method effectively relieves the impact of uneven illumination and improves the performance of feature extraction for blurred images. Furthermore, our proposed ISLS method obtains detailed leakage areas, further helping technical people to analyze the reasons for leakage and adjust sauce-filling machines (e.g., sauce nozzles, high-temperature sealing devices, and shaft rotation speeds).

Our main contributions can be summarized as follows:(1)To alleviate blurred image issues caused by uneven illumination, we propose the ULIE method via an illumination-aware mechanism to enhance the texture details of leakage within the ROI.(2)The MFFM is proposed to fuse multi-level features with a small number of parameters, capturing multi-scale features to effectively relieve the issue of missing information.(3)To alleviate the interference of invalid information, we introduce the SSAM by combining spatial and channel attention mechanisms to enhance the discriminability of valid features in the ROI.(4)We generate a sauce-packet dataset to facilitate research. Furthermore, our method, Mean Intersection over Union (mIoU), achieves 80.8% and Mean Pixel Accuracy (mPA) reaches 90.1% on the self-built dataset, which are +0.9% and +0.9% higher than the previous CNN method [23].

The rest of this paper is organized as follows: Section 2 describes the proposed method in detail. Section 3 conducts the experiments based on the self-built dataset to verify the effectiveness of the proposed method. Section 4 summarizes this paper.

## 2. Materials and Methods

### 2.1. Overall Architecture

We propose illumination-aware sauce-packet leakage segmentation (ISLS) for the industrial production line, as illustrated in Figure 1. The ISLS method includes an NVIDIA GPU and an intelligent sensor with a hardware accelerated EdgeAI development environment. We train the model on NVIDIA GPU and perform inference on intelligent sensors.

The cameras capture images of sauce-packets and transfer the images to NVIDIA GPU and intelligent sensors, where the images are stored as a raw dataset. The size of the raw dataset is 888 × 1112 (height and width), comprising 606 sauce-packet leakage images with varying degrees and shapes. The sauce-packet dataset is pre-processed using the YOLO-Fastestv2 [22] detection algorithm to focus on the Region of Interest (ROI) and is augmented by our proposed uneven-light image-enhancement (ULIE) method. We split the raw dataset into training and validation sets in a 7:3 ratio for second-stage segmentation. Notably, the input image size for ISLS is 888 × 1112, and the output size is 128 × 512.

The overall ISLS flow and the image size changes at different ISLS stages are shown in Figure 2, providing a comprehensive view of the process from raw dataset acquisition to segmentation.

We train our proposed leakage segmentation network on the NVIDIA GPU. Subsequently, we assess the robustness and generalization of our trained model by cross-validation. The trained model is deployed on the NVIDIA Jetson TX2 intelligent sensors for inference, where it is utilized to identify real-time leakage in images (i.e., highlight the areas of leakage with bounding boxes).

We first introduce uneven-light image enhancement in our designed ISLS, including the localization and ULIE method. Subsequently, we present the segmentation network, proposing the multi-scale feature fusion module (MFFM) and the Sequential Self-Attention Mechanisms (SSAMs). Detailed explanations of these components will be presented in subsequent sections.

### 2.2. Uneven-Light Image Enhancement for Illumination-Aware Region Enhancement

The actual leakage segmentation of sauce packets is often influenced by uneven light sources, which consist of insufficient illumination and overexposure. To relieve the problem of image blurring caused by uneven illumination, we propose the uneven-light image-enhancement (ULIE) method, employed in the illumination-aware region-enhancement stage of ISLS. The ULIE method is inspired by the relevant image-enhancement algorithms [24,25,26]. Our ULIE method can enhance the illumination of sauce-packet images under insufficient illumination conditions and improve the image contrast and texture details under overexposure conditions.

The input of ISLS is in a three-channel RGB format, where R, G, and B represent the color space values of red, green, and blue, respectively. We utilize the mean function in OpenCV to calculate the mean value of the RGB three channels in the ROI. Through extensive experimental analysis, we define 115 and 180 as the thresholds for insufficient illumination and overexposure, respectively. The implementation details of our ULIE method are as follows:

In the case of insufficient illumination, the ULIE method is built upon the retinex model [27,28]. The retinex model theory posits that a color image can be decomposed into two primary components: the illumination component (lighting) and the reflection component, as shown in Equation (1).
Li(x) = Tr(x) ◦ Re(x)(1)
where Li(x) and Re(x) represent the input image and the image to be recovered, respectively. Tr(x) represents the illumination-mapping image, and the ◦ operator represents the element-wise multiplication.

Firstly, to simplify the computation of the ULIE method, it is commonly assumed that the three channels of images share the same illumination map [29]. The ULIE method calculates the maximum value among the RGB channels of the image to independently estimate the illumination of each pixel x, obtaining the initial estimation:(2)$$\mathrm{Tr}(x)=\underset{c\in \{R,G,B\}}{\max}Li^{c}\left(x\right)$$
where x represents individual pixel, c represents channels, and $Li^{c}\left(x\right)$ is the input image of the maximum channel in the RGB.

Secondly, to ensure that the illumination map does not cause the enhanced image to become overly saturated, the ULIE method modifies Re(x):(3)Re(x)=LixTrx+ϵ
where ϵ is a very small constant, to avoid denominating to zero.

Thirdly, the ULIE method employs the augmented Lagrangian multiplier optimization method [30] to preserve the structural information and smooth texture details of sauce-packet images. The ULIE method introduces the following optimization problem to accelerate the processing speed of sauce-packet images:(4)$$\underset{\mathrm{Tr}}{\min}{\left||\hat{Tr}-Tr|\right|}_{F}^{2}+\alpha {\left||W\circ ▽Tr|\right|}_{1}$$
where ||·${\left|\right|}_{F}$ and ||·${\left|\right|}_{1}$ represent the F norm and L1 regularization, and α is the coefficient balancing the F norm and L1 regularization, respectively. Additionally, W is the weight matrix, and ▽Tr represents a first-order derivative filter, encompassing both horizontal and vertical directions.

Finally, the ULIE method iteratively updates according to the retinex model, being solved to obtain the result image Re(x) in Equation (1). The ULIE method applies BM3D [31] for the denoising optimization of the result image Re(x). To reduce the computation of the denoising process in ULIE, the method transforms the RGB three channels of the result image Re(x) into YUV three channels [32] and performs denoising only on the Y channel:Y = 0.299R + 0.587G + 0.114BU = −0.169R − 0.331G + 0.5B + 128V = 0.5R − 0.419G − 0.081B + 128(5)
where Y represents luminance, and U and V represent blue chrominance and red chrominance, respectively.

In the case of overexposure, the ULIE method divides the image into blocks to obtain overexposure regions (i.e., locally overexposed areas). Firstly, to obtain the illumination information of the ROI, we convert the RGB color space into the YUV color space, as shown in Equation (5). The ULIE method divides the input image into several small blocks and performs Contrast-Limited Adaptive Histogram Equalization (CLAHE) [33] on each block to enhance the clarity of the image. CLAHE clips and redistributes the histograms of each sub-image, thereby limiting the degree of contrast enhancement. CLAHE prevents the amplification of noise and excessive enhancement [34]. The ULIE method initially divides the original image into several non-overlapping sub-images, each denoted as s. We compute the frequency of pixel values $p_{s}\left(i\right)$, representing the data distribution of pixel values i within each sub-image. The definition of $p_{s}\left(i\right)$ is given by Equation (6):(6)$$p_{s}\left(i\right)=\frac{n_{i}}{N}$$
where $p_{s}\left(i\right)$ represents the frequency of pixel values equal to i, $n_{i}$ represents the number of pixels with a pixel value of i, and N represents the total number of pixels in the sub-image.

Secondly, the ULIE method computes the Cumulative Distribution Function (CDF) $c_{s}\left(i\right)$ for each sub-image s in Equation (7), representing the cumulative frequency of pixel values less than or equal to i:(7)$$c_{s}\left(i\right)=\sum_{j=0}^{i}p_{s}\left(j\right)$$
where $c_{s}\left(i\right)$ represents the CDF for the pixel value i, and $p_{s}\left(j\right)$ represents the frequency of pixel values equal to j.

Thirdly, the ULIE method utilizes Equation (8) to compute the transformation function $T_{s}\left(i\right)$ for each sub-image s, representing the function that maps the original pixel value i to a new pixel value:(8)$$T_{s}\left(i\right)={\lfloor c}_{s}\left(i\right)\rfloor$$
where $T_{s}\left(i\right)$ represents the transformed pixel value for the original pixel value i, L represents the maximum range of pixel values, and ⌊·⌋ represents the floor function.

The ULIE method clips and redistributes $p_{s}\left(i\right)$ for each sub-image s, limiting the degree of contrast enhancement, which prevents the amplification of noise and excessive enhancement [33]. Finally, the ULIE method consolidates all transformed sub-images $T_{s}\left(i\right)$ into the final image and converts the image from YUV format back to RGB format.

The results of ULIE images are shown in Figure 3, where the left image is the non-optimized image, and the right image is the optimized image. Figure 3a shows that the image has improved overall illumination, with a clearer boundary between the leakage and the background. The ULIE method effectively enhances the image contrast and clarity. Figure 3b reveals that the illumination of the optimized image is more balanced. The ULEIE method alleviates the phenomenon of local overexposure, which further proves that our method effectively avoids gray jump [35]. We perform convolution and downsampling operations on the yellow and red box regions, obtaining the corresponding feature maps between non-optimized and optimized images. It is shown that the details of the optimized feature map are more obvious. In summary, through the above process, our method can effectively enhance the details and textures of sauce-packet images under insufficient illumination and overexposure.

### 2.3. ISLS Network Details for Leakage Segmentation

In the leakage region segmentation stage of the ISLS method, we propose our network with the EdgeNext backbone, which comprises only 1.3 M parameters [36]. The EdgeNext integrates the advantages of the Convolutional Neural Network (CNN) and vision transformer (ViT). The CNN extracts local features of images using convolution operation [37], and the ViT [38] captures global contextual information of images. The network is an end-to-end network, where the input channel dimension is 3 (i.e., RGB), and the input image size is 128 × 512. Our overall network structure is shown in Figure 4, which includes the encoder and decoder.

The encoder fuses the local and global representation. Firstly, the n × n Conv encoder consists of three modules. The n × n Conv encoder utilizes adaptive kernels to adjust the size of convolutional kernels based on distinct network layers, which aims to decrease computational complexity and enhance the receptive field [39]. Secondly, the SDTA encoder combines spatial and channel information. The SDTA encoder utilizes deep transposed convolution and adaptive attention mechanisms, which improve the performance of capturing local and global representation. Thirdly, the information of deep- and shallow-layer feature maps is fused by our MFFM, which improves the performance of encoder feature extraction. Our MFFM structure is shown in Figure 5.

Specifically, we extract four feature maps of different sizes from the encoder, denoted as $x_{1}$, $x_{2}$, $x_{3}$, and $x_{4}$. Firstly, the MFFM adjusts the channel number of the feature map $x_{1}$ to 512 through a 1 × 1 convolution and 4× downsampling. Next, the MFFM applies similar operations with $x_{1}$ to the feature map $x_{2}$, with 2× downsampling, as illustrated in Equation (9):(9)$$x_{i}^{’}=\mathrm{MaxPooling}\left(\mathrm{Conv}\left(x_{i}\right)\right),\;i\in \left\{1,\;2\right\}$$
where MaxPooling represents the downsampling process through maximum-pooling operation, and Conv represents the convolution operator.

Secondly, $x_{3}$ has the same size as the output. Therefore, the MFFM only needs to utilize a 1 × 1 convolution to adjust the channel number of the feature map $x_{3}$ to 512, as shown in Equation (10):(10)$$x_{3}^{’}=\mathrm{Conv}\left(x_{3}\right)$$

Thirdly, the channel number of $x_{4}$ is same with the output; therefore, the MFFM only performs an ${\mathrm{Upsampling}}_{2}$ operation on the feature map $x_{4}$ ${\mathrm{Upsampling}}_{2}$ is achieved using nearest-neighbor interpolation, as depicted in Equation (11):(11)$$x_{4}^{’}={\mathrm{Upsampling}}_{2}(x_{4})$$
where ${\mathrm{Upsampling}}_{2}$ represents 2× upsampling operation.

Through the above operations, the feature maps $x_{1}^{’}$, $x_{2}^{’}$, $x_{3}^{’}$, and $x_{4}^{’}$ are obtained. Finally, we fuse the feature information of $x_{1}^{’}$, $x_{2}^{’}$, $x_{3}^{’}$, and $x_{4}^{’}$ to output the feature map $x_{5}$, as shown in Equation (12):(12)$$x_{5}=\sum_{i=1}^{4}x_{i}$$

The reasons for the MFFM’s small parameter number is that the Conv operator employs a 1 × 1 convolution, the ${\mathrm{Upsampling}}_{2}$ operation uses nearest-neighbor interpolation, and downsampling is achieved through Maxpooling. The 1 × 1 convolution operation only increases a small number of parameters, and upsampling and downsampling operations do not increase the number of parameters. Compared to the Feature Pyramid Network (FPN) [40] and AF-FPN [41], the parameter number of our MFFM is relatively small. That is, our MFFM has a parameter number of only 0.23 M, with only 3.05% of the FPN and AF-FPN parameter number.

The decoder includes a skip connection and the SSAM. The SSAM in stage 1 of the decoder aims to improve the identification of salient features. The SSAM keeps a high resolution in both channel and spatial branches, which enhances the salient features of the sauce-packet ROI. Specifically, the SSAM contains two modules, consisting of a channel-only module and the spatial-only module, as shown in Figure 6. For the channel-only module of the SSAM, the output $y_{\mathrm{in}}$ is generated by fusing the feature map $x_{\mathrm{in}}$, obtained from both the skip connection and the channel attention mechanism. The specific computational process for the channel attention mechanism is shown in Equation (13).
(13)ych=SReConvxin⊗ReConvxinych’=SigLNConvych
where $x_{\mathrm{in}}$ and $y_{\mathrm{ch}}$ represent the input and output of the SSAM channel-only module. Conv represents the convolution operator, Re represents reshape operator, and S represents the softmax operator. Additionally, LN represents layer normalization, and Sig represents the sigmoid function.

The process of the SSAM spatial-only module is similar, with one part from the skip connection, and the other part from the spatial attention mechanism, as shown in Equation (14) for specific operations:(14)ysp=SReGPConvyin⊗ReConvyinyout=yin⊙SigLNysp
where $y_{\mathrm{in}}$ and $y_{\mathrm{out}}$ represent the input and output of the SSAM spatial-only module. GP, S, and LN represent to the global pooling operator, softmax operator, and layer normalization, respectively. ⊗ and ⊙ represent the tensor product and multiplication operations, respectively.

Stages 2 to 5 of our proposed network decoder contain the SSAM and Concat operator. The Concat operator concatenates feature maps of two branches in the channel dimension, as shown in green box of Figure 3. Specifically, the one-branch feature map comes from the SSAM output, which is upsampled 2×. The other branch feature map comes from skip connections, which can avoid gradient vanishing and improve the training speed of the network [42].

## 3. Experiments and Results

### 3.1. Dataset and Experiment Setting

Currently, there are almost no publicly available datasets for sauce-packet leakage segmentation. Therefore, we generated a dataset at Nanjing University of Posts and Telecommunications, captured by an industrial high-speed camera, namely the MER2-134-90GC, and the Daheng Image industrial lens HN-P-1628-6M-C2/3. Our self-built dataset comprises images with varying degrees of leakage. Specifically, it includes 315 images under normal illumination conditions, 143 images under insufficient illumination, and 148 images with overexposure. Some examples from the self-built dataset are presented in Figure 7.

The backbone of our proposed network is EdgeNext. Our experimental environment is PyTorch 2.0.1. We divide the dataset into training and validation sets, containing 424 and 182 images, respectively. During the training phase, we employ the NVIDIA RTX GPU 4060, while for inference, we deploy it to the NVIDIA Jetson TX2. Meanwhile, we utilize the cross-validation strategy to verify the robustness of the model. During training, we use DiceLoss [43] to measure the degree of similarity between the predicted results and the ground truth. In addition, we employ FocalLoss [44] to relieve class imbalance by emphasizing hard-to-classify examples, which makes the model pay more attention to challenging pixels. During deployment, we utilize pruning technology to accelerate our model.

### 3.2. Evaluation Indexes

To evaluate the performance of the method, we selected seven widely used evaluation indices: Average Precision (AP), Mean Intersection over Union (mIoU), Mean Pixel Accuracy (mPA), F1-Score, Params, Giga Floating Point Operations Per Second (GFLOPs), and Frames per Second (FPS). AP is an evaluation index for YOLO-Fastestv2, with higher values indicating stronger performance. Params is employed as the evaluation index for the model parameters. GFLOPs is an index measuring a model’s computational complexity. And FPS is the evaluation index for the inference speed of a model. The definitions are shown in Equations (15)–(18):(15)$$\mathrm{mIoU}=\frac{1}{n}\sum_{i=1}^{n}\frac{\mathrm{TP}}{\mathrm{TP}+\mathrm{FP}+\mathrm{FN}}$$
(16)$$\mathrm{mPA}=\frac{1}{n}\sum_{i=1}^{n}\frac{\mathrm{TP}}{\mathrm{TP}+\mathrm{FP}}$$
(17)$$F1-\mathrm{score}=\frac{2\times \mathrm{precision}\times \mathrm{recall}}{\mathrm{precision}+\mathrm{recall}}$$
(18)$$\mathrm{AP}=\int_{0}^{1}P\left(R\right)dR$$
where precision and recall represent TP/(TP + FP) and TP/(TP + FN), respectively, and P(R) is the precision at a given recall rate R.

### 3.3. YOLO-Faststv2 Training

Segmenting the entire image increases the computational burden, and it is challenging to differentiate between the connection and the main sauce, as shown in Figure 6, often resulting in incorrect segmentation. Therefore, during the illumination-aware region-enhancement stage of the ISLS, we employ YOLO-Fastestv2 for ROI cropping. The dataset for YOLO-Fastestv2 is divided into a training set and a validation set with a ratio of 7:3. The input image size for YOLO-Fastestv2 is set at 888 × 1112, while the output image dimensions are resized to 128 × 512. The ROI images are enhanced by the ULIE method and inputted into the leakage region segmentation stage of the ISLS. The training process and evaluation indexes of YOLO-Fastestv2 are illustrated in Figure 8. Significantly, YOLO-Fastestv2 attained the AP of 99.95%, which can primarily be attributed to the simple and uniform features of the sauce-packet connections.

### 3.4. Experiment Analysis of ULIE

We employ the grid search method to explore the optimal illumination thresholds of ULIE for insufficient illumination and overexposure. And the controlled variable method is utilized to reflect trends in performance. In the case of insufficient illumination, to validate the optimal threshold as 115, we maintain the overexposure threshold at 180. We adjust the insufficient illumination threshold within the range of 75 to 145. The ISLS evaluation indices (i.e., mIoU, mPA, Accuracy, F1-score) for different thresholds are shown in Figure 9. We observe that the evaluation indices reach their optimum when the threshold is set at 115.

Simultaneously, to verify the optimal threshold for overexposure as 180, we maintain the insufficient illumination threshold at 115. We adjust the overexposure threshold within the range of 150 to 220. The ISLS evaluation indices for various thresholds are presented in Figure 10. It is noted that the evaluation indices achieve their peak performance when the threshold is set at 180.

To verify the effectiveness of our proposed ULIE method, in addition to the aforementioned quantitative analysis, we also conduct the qualitative analysis. Figure 11 shows the qualitative segmentation results with and without using the ULIE method.

In summary, we define the thresholds of insufficient illumination and overexposure as 115 and 180, respectively.

### 3.5. Analysis of Ablation Study

In this paper, we propose the two stage illumination-aware sauce-packet leakage segmentation (ISLS) method. The ISLS method consists of the uneven-light image-enhancement (ULIE) method, the multi-scale feature fusion module (MFFM), and the Sequential Self-Attention Mechanism (SSAM). To assess the performance of ISLS, we conducted ablation studies on each component.

Ablation for backbone: Our baseline employs a U-shaped segmentation network that selects the EdgeNext feature extraction network as its backbone. To verify the superiority of EdgeNext as the backbone network for the U-shaped segmentation network, we compare the performances of MobileViTv2 [45], MobileNetv3 [46], MobileOneS4 [47], Regnetx [48], and EfficientNetv2s [49] as U-shaped segmentation networks’ backbones, as shown in Table 1. When compared with MobileOneS4, our baseline achieves the mIoU of 75.6% (+0.3%) and the mPA of 85.6% (+2.0%), respectively. The parameters and GFLOPs of EdgeNext are close to those of MobileViTv2 and MobileNetv3. And the parameters of EdgeNext are only half of MobileOneS4 and EfficientNetv2s, with its GFLOPs being inferior to both MobileOneS4 and EfficientNetv2s. MobileViTv2 employs lightweight transformer modules to delve into the intricate relationships between leakage features, encountering challenges in capturing the subtle textures and minor discrepancies in leakage representation. MobileNetv3 and MobileOneS4 utilize depthwise separable convolutions to reduce computational demands, which constrains the interaction capabilities among features across different channels, resulting in an insufficient representation of leakage. EdgeNext is adept at capturing the details and textures within images, which is crucial when dealing with sauce packets that feature intricate textures and complex backgrounds. Hence, the network architecture of EdgeNext is more suited to capturing the features of sauce-packet leakage.

Ablation for ULIE: We propose ULIE to alleviate the problem of blurring images in uneven illumination conditions. In Table 1, compared to the baseline, mIoU improves by about 1.5% and mPA by about 3.9% after using ULIE. R2RNet [21] alleviates low-light image degradation but struggles with blurring from uneven illumination on sauce packets, failing to retain leakage texture. Moreover, the R2RNet’s high computational demand makes it hard to satisfy real-time detection. The ULIE method enhances the visibility of subtle leakage details by effectively boosting the local contrast and illumination of images, thereby significantly improving segmentation performance. The ULIE method is not based on deep learning, which relies on the adjustment and optimization of image illumination and contrast. Hence, the ULIE method does not require any parameters to be trained. Through Section 2.2, Methods and Formulas, the ULIE method adaptively enhances the illumination and contrast of images.

Ablation for MFFM: The mIoU and mPA of the MFFM reach 78.7% and 87.9%, as shown in Table 1. Compared with the baseline, the mIoU and mPA of the MFFM increase by 3.1% and 2.3%. In comparison to other feature fusion modules, the MFFM exhibits significantly fewer parameters than both the FPN and AF-FPN, with only 3.05% of those of the FPN and AF-FPN. Additionally, the mIoU and mPA of the MFFM are 1.4% and 2.7% higher than the FPN, and the mIoU and mPA of the MFFM are 3.4% and 4.3% higher than the AF-FPN. Significantly, the FPN and AF-FPN structures are complex, and the GFLOPs of the FPN and AF-FPN reach as high as 1655. However, the features of leakage are monotonous, rendering the extraction of salient features less effective with the FPN and AF-FPN. Compared to the FPN and AF-FPN, our proposed MFFM employs a more efficient feature-integration mechanism, with only 30 GFLOPs.

Ablation for SSAM: We add the attention mechanism to ISLS to improve the performance of feature extraction and compare it with Global Attention Mechanism (GAM) [50] and the Simple, Parameter-Free Attention Module (SimAM) [51]. In Table 2, it can be observed that utilizing the SSAM results in an improvement of about 2.4% in mIoU and 0.9% in mPA with the GAM. The GAM distributes attention weights across a global scope with plenty of global pooling and fully connected layers, leading to high computational complexity and failing to capture detailed local features adequately. The GFLOPs of the SSAM are only 3.237, significantly lower than the approximately 52.613 GFLOPs for the GAM. Additionally, Table 2 shows that BUM with the SSAM improves the values by about 3.7% in mIoU and 3.5% in mPA compared to BUM with the SimAM. Although the SimAM does not increase the computational load and inference time, the SimAM exhibits limited performance when addressing complex and subtle leakage features. In summary, our introduced SSAM enhances the identification of salient features, which improves the focus on details and textures of leakage.

To further analyze the effectiveness of our method, we use the Grad-cam [53] to visualize the attention heatmap. As shown in Figure 12, the redder the heatmap, the more the attention mechanism focuses on the feature.

Figure 12c exhibits strong capabilities for global feature extraction but shows insufficient performance in capturing local information. Figure 12b,d do not fully focus on the leakage regions, and the problem of attention divergence exists. Our ISLS method with the SSAM effectively alleviates interference from invalid information, which results in more focus on leakage regions, as shown in Figure 12e.

The leakage segmentation results of sauce packets with different attention mechanisms are shown in Figure 13. Our method with the SSAM achieves more refined leakage boundary segmentation for sauce packets by employing spatial- and channel-adaptive weighting.

Our proposed ISLS achieves 4 FPS on the NVIDIA Jetson TX2. Our ISLS meets the real-time requirements for industrial applications, which demand a minimum of 3 FPS. Figure 14a illustrates the CPU utilization of the ISLS method. The GPU utilization of the leakage segmentation stage is illustrated in Figure 14b.

Before optimization, the GPU utilization often approached 100% (green line), posing risks of system crashes or failures. To ensure system stability and device reliability, we optimized the ISLS method by integrating L1 unstructured pruning technology [54], with a reduction of 20% in the model’s parameter. Performing fine-tuning training while pruning can effectively mitigate the performance degradation caused by pruning, with almost no decrease in accuracy. The GPU utilization after optimization decreases to a certain extent (red line), accelerating model inference while simultaneously decreasing GPU load and enhancing system stability.

Figure 14c illustrates the comparison between the FPS before optimization (green line) and after optimization (red line). The experimental results demonstrate a significant improvement in the performance of our method after optimization. Compared with the unoptimized method, the optimized method improves the FPS by 2.7 times.

Experimental results demonstrate that ISLS performs well under uneven illumination conditions. We deploy the ISLS on intelligent sensors, and the system fulfills the real-time requirements of industrial applications.

### 3.6. Comparison with Other Segmentation Methods

In this section, we compare ISLS with several state-of-the-art (SOTA) semantic segmentation networks, including HRNet [13], BiseNetv2 [14], SegFormer [15], PSPNet [55], DeepLabv3 [56], and LIEPNet [23]. The evaluation results are shown in Table 3. The design advantage of our model alleviates the negative impact of uneven illumination and effectively captures semantic features of multi-level leakage. Consequently, we achieve the highest accuracy, with mIoU, mPA, and F1-score reaching 80.8%, 90.1%, and 88.8%, respectively.

In the task of sauce-packet leakage detection, other semantic segmentation networks such as HRNet, BiseNetv2, SegFormer, PSPNet, and DeepLabv3 exhibit limited generalization performance in the current field. The limited generalization performance is due to the low contrast and unclear textures of leakage under uneven illumination conditions. Additionally, sauce-packet leakage detection demands the capability of networks to handle features across various scales to identify leakages ranging from small to large sizes and shapes. Although HRNet and PSPNet have multi-scale feature fusion mechanisms, their capability to extract monotonous leakage features is limited. The generalization performance of HRNet and PSPNet is insufficient under uneven illumination conditions.

For practical production line applications, detection speed is crucial. Networks like DeepLabv3 and SegFormer, despite their high accuracy, have high computational demands, making them unsuitable for real-time scenarios. After multiple downsampling and pooling operations, LIEPNet leads to a loss in spatial representation. Meanwhile, LIEPNet employs the SimAM module, which encounters difficulties in effectively distinguishing between the foreground and background under uneven illumination conditions.

Additionally, we compare ISLS with several classical traditional segmentation methods, including template matching [57], Canny edge segmentation [58], contour segmentation [59], PCA segmentation [60], and iForest segmentation [18]. The evaluation results of traditional leakage segmentation algorithms are shown in Table 4.

The performance of traditional methods is insufficient in the sauce-packet leakage segmentation, as shown in Figure 15. On the one hand, some traditional algorithms mistakenly identify sealing imprints on the sauce packet and the black blocks, as shown in Figure 16. On the other hand, these algorithms are significantly affected by uneven lighting conditions. These above problems of traditional algorithms result in lower accuracy in sauce-packet leakage segmentation.

### 3.7. Generalization Performance Validation

To further verify the generalizability of our proposed ISLS method, we compare ISLS with CNN methods on the Medical Dermoscopic Image dataset ISIC [62] and the rail surface defect dataset RSDDs [63]. The experimental results on the ISIC dataset are shown in Table 5, and the results on the RSDDs dataset are presented in Table 6. Our ISLS method exhibits certain advantages in the current scenario. For example, ISLS achieves an AUC of 93.2% on the ISIC dataset and 88.4% on the RSDDs dataset, exceeding the L-SVM [62] by 0.6% and NDD-Net [64] by 0.2%, respectively. Although CCEANN [65] achieves an F1-score of 92.0%, its parameter size is as high as 167.28 M, which far exceeds our ISLS method. Consequently, CCEANN fails to satisfy the real-time processing demands of edge sensor devices.

These improvements can be attributed to several key factors: (1) The first-stage ULIE of the ISLS method effectively reduces noise caused by insufficient illumination and mitigates the blurring due to overexposure. The ULIE method can effectively extract discriminative features in images under uneven illumination conditions, enhancing the feature and texture details. (2) Our MFFM module efficiently integrates features from different levels, capturing texture features while preserving edge detail. (3) Introducing the SSAM module within the ISLS method improves the representation capabilities in both the channel and spatial dimensions, resulting in significant performance enhancements.

### 3.8. Discussion

The results presented above show that our proposed ISLS method is essential for blurring sauce-packet leakage image segmentation.

The image classification CNN model lacks the capacity to precisely localize leakage areas and quantify the extent of leakage. The traditional image-processing algorithms make it hard to capture the representation of leakage under uneven illumination conditions, leading to the mistaken identification of sealing imprints on sauce packets and black blocks. The object location algorithms (e.g., YOLO) struggle to accurately capture information regarding the area and shape of irregular leakage regions. For the current task, employing detection algorithms to label irregular leakage is unreasonable and fails to capture specific leak areas accurately.

However, our ISLS method alleviates the above problems. Our ISLS method consists of the illumination-aware region-enhancement stage and the leakage region segmentation stage. The first stage of the ISLS method reduces the computational load and alleviates the image blurring problem. The second stage enhances the ability to capture leakage through the MFFM, SSAM, and our proposed U-shaped network. Our MFFM shows superior capability in capturing monotonous leakage features, while its computational load is significantly lower than that of FPN and AF-FPN. Our SSAM improves the ability to focus on the details and textures of the leakage.

Furthermore, to verify our ISLS method, we compare various algorithms’ performance with ISLS on medical dermoscopic image datasets and rail surface defect datasets. Our ISLS achieves the best balance between accuracy and speed. These datasets have similar blurring problems from uneven illumination. Our ULIE method reduces image noise under insufficient illumination, avoids the gray jump phenomenon under overexposure, and improves feature capture capability in uneven lighting conditions.

Overall, our proposed two-stage ISLS method achieves strong performance on the blurring sauce-packet leakage segmentation task and is applicable to other general domains. Furthermore, we built a sauce-packet leakage dataset, which is the first dataset in the BSLST. The ISLS method alleviates the leakage detection of transparent sauce packets where uneven illumination is caused by shaking. For opaque-material packets, other morphological recognition methods are needed for leakage detection, which will be our future research.

In addition, we will further focus on model parameter compression and precision tuning. Meanwhile, we will contemplate the deployment of the state-of-the-art SAM large model on the edge to augment the quality inspection rate of industrial sauce-packet products, thereby enhancing the efficacy of quality control processes.

## 4. Conclusions

In this paper, our objective is to address the issue of detecting leakage in blurred images under uneven illumination conditions. We propose the illumination-aware sauce-packet leakage segmentation (ISLS) method, consisting of illumination-aware region enhancement and leakage region segmentation stages. The first stage of ISLS reduces redundant computations of image enhancement processing and alleviates the image blurring caused by uneven illumination, which effectively enhances image details and textures. In the second stage of ISLS, we design a leakage segmentation network. In our proposed network, the multi-scale feature fusion module (MFFM) efficiently fuses the shallow- and deep-layer features with a small number of parameters, which improves the feature extraction performance. Additionally, the Sequential Self-Attention Mechanism (SSAM) achieves feature enhancement in both channel and spatial dimensions, improving the identification of salient features. Extensive experiments on our self-built dataset demonstrate that our method effectively alleviates the blurred sauce-packet imaging issue and outperforms existing algorithms. And the generalizability experiments on the ISIC and RSDDs datasets illustrate that our method possesses certain advantages. Furthermore, our method improves the stability and reliability of industrial systems and reduces the waste of production resources. The performance testing of the intelligent sensors also validates the suitability of our ISLS method for the current scenario.

## Figures and Tables

**Figure 1 sensors-24-03216-f001:**
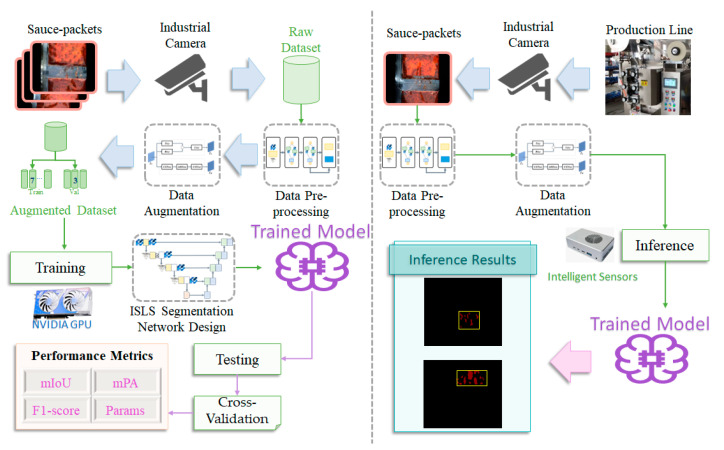
The data-processing procedure of ISLS including training (**left**) and inference stages (**right**).

**Figure 2 sensors-24-03216-f002:**
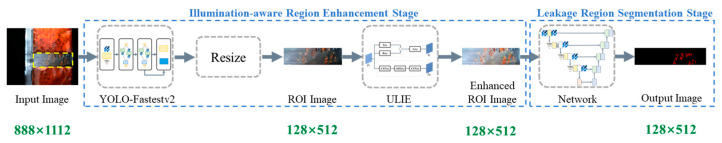
The overall ISLS flow.

**Figure 3 sensors-24-03216-f003:**
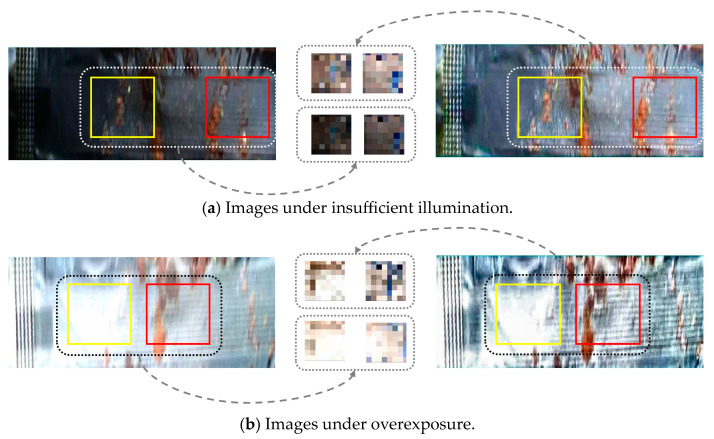
The visual results of sauce packets under (**a**) insufficient illumination and (**b**) overexposure conditions. The left side represents non-optimized images, the right side represents optimized images, and the central part represents feature maps between non-optimized and optimized images.

**Figure 4 sensors-24-03216-f004:**
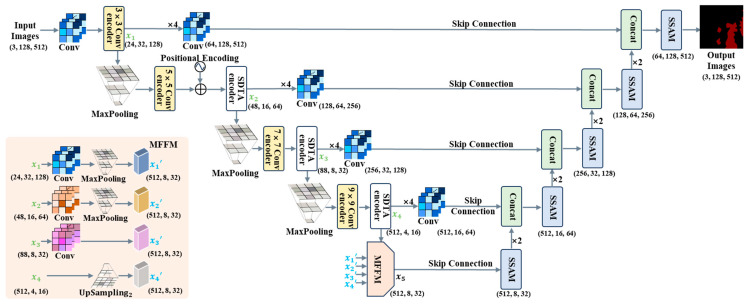
The overall network structure of the ISLS segmentation stage, which consists of an encoder and decoder. The input image format follows the structure of (channels, height, width).

**Figure 5 sensors-24-03216-f005:**
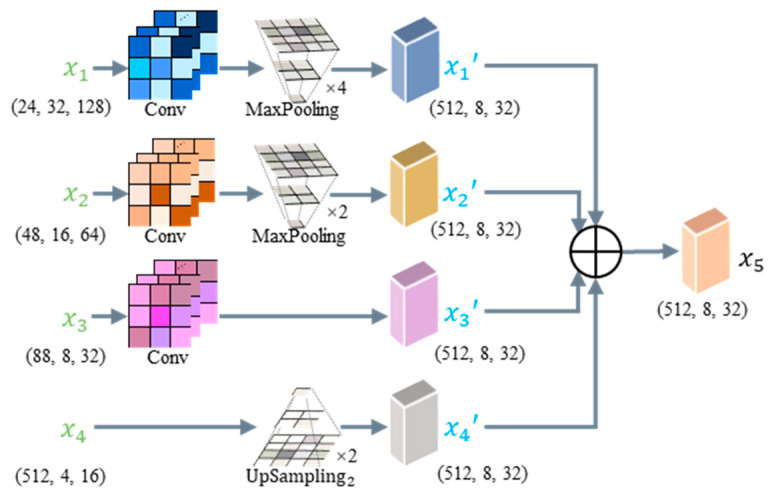
The overall MFFM structure. The MFFM fuses information from four levels, with varying feature sizes for each layer. The size of feature maps is (C, H, W), where C, H, and W represent channel dimension, image height, and image width, respectively.

**Figure 6 sensors-24-03216-f006:**
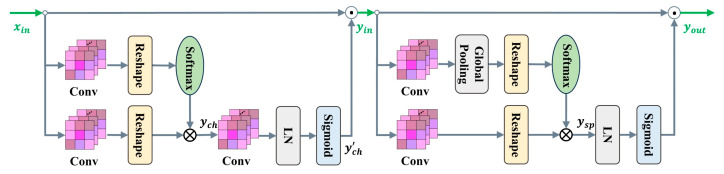
Sequential Self-Attention Mechanism (SSAM), which includes the channel-only branch (**left**) and the spatial-only branch (**right**).

**Figure 7 sensors-24-03216-f007:**
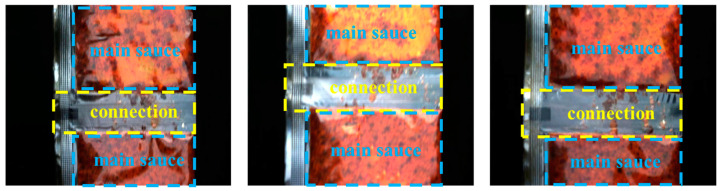
Partial sample of sauce-packet dataset. The yellow dotted boxes indicate the connection of sauce-packet (i.e., ROI), and the blue dotted boxes indicate the main sauce.

**Figure 8 sensors-24-03216-f008:**
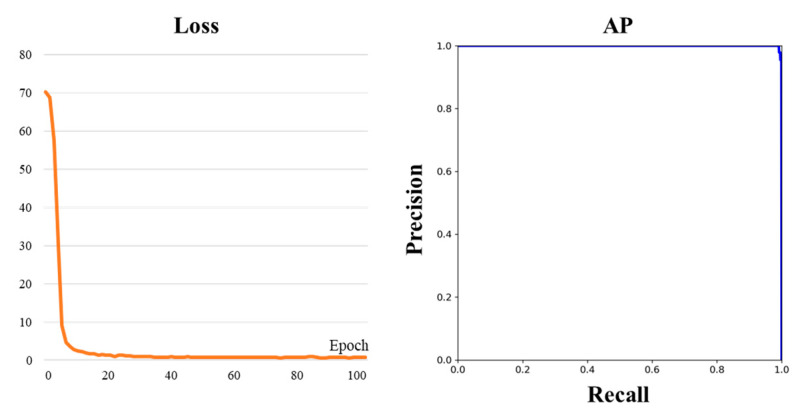
Loss variation curve during the training process and validation results of YOLO-Fastestv2.

**Figure 9 sensors-24-03216-f009:**
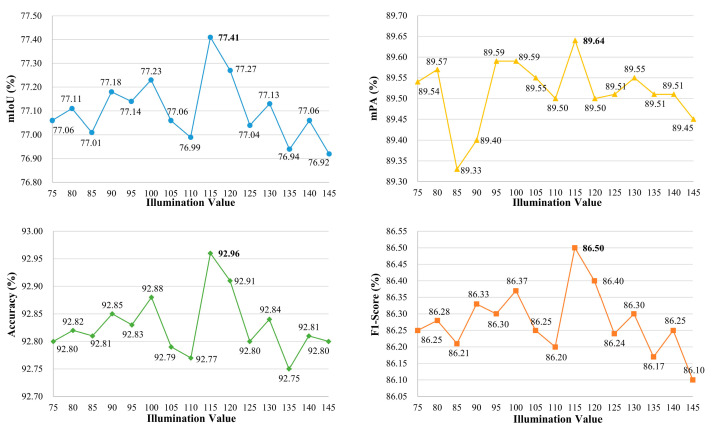
The evaluation indexes under different insufficient illumination thresholds, with a fixed overexposure threshold of 180.

**Figure 10 sensors-24-03216-f010:**
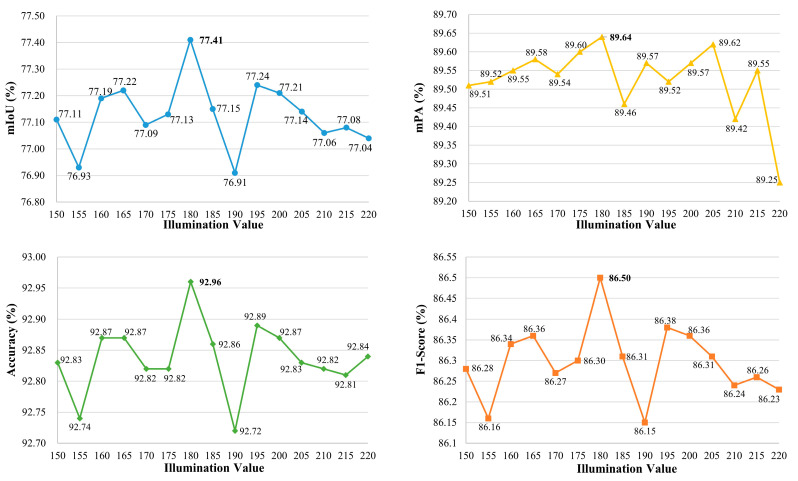
The evaluation indexes under different overexposure thresholds, with a fixed insufficient illumination threshold of 115.

**Figure 11 sensors-24-03216-f011:**
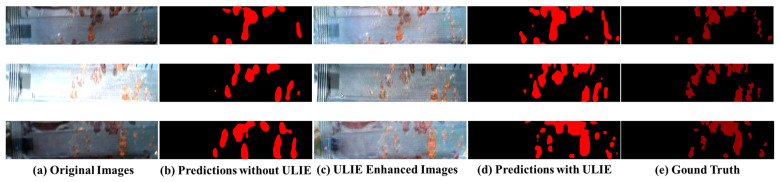
The segmentation results with and without the ULIE method.

**Figure 12 sensors-24-03216-f012:**
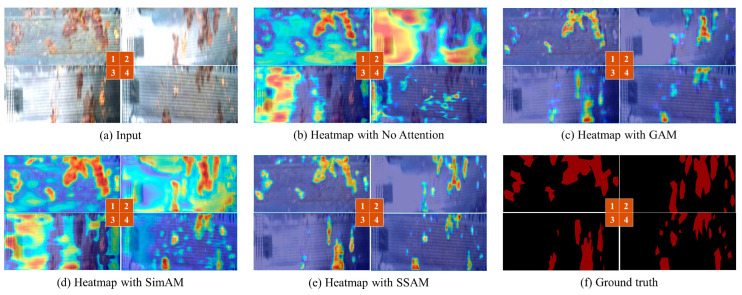
Heatmap results using different attention mechanisms.

**Figure 13 sensors-24-03216-f013:**
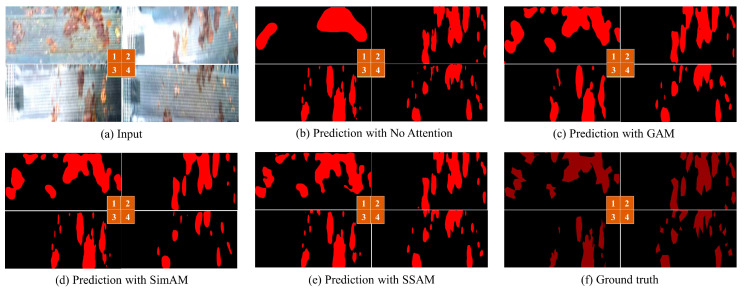
Prediction results using different attention mechanisms.

**Figure 14 sensors-24-03216-f014:**
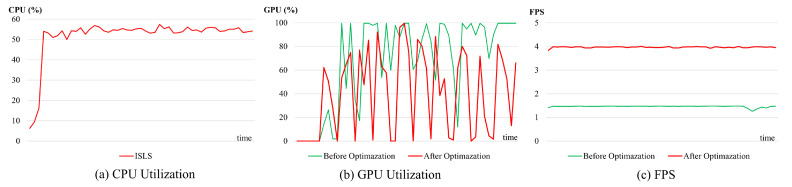
Relevant performance statistics of ISLS method during inference: (**a**) the CPU utilization of our method after optimization, (**b**) a comparison of GPU utilization before and after optimization, (**c**) a comparison of FPS changes before and after optimization.

**Figure 15 sensors-24-03216-f015:**
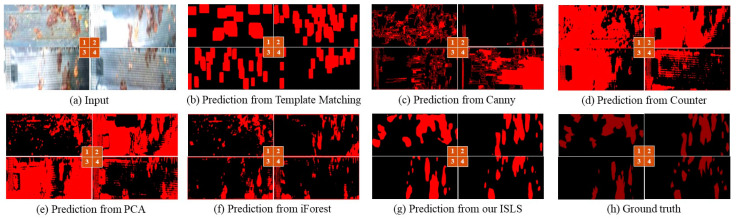
Comparison of prediction results between various traditional algorithms and our ISLS.

**Figure 16 sensors-24-03216-f016:**
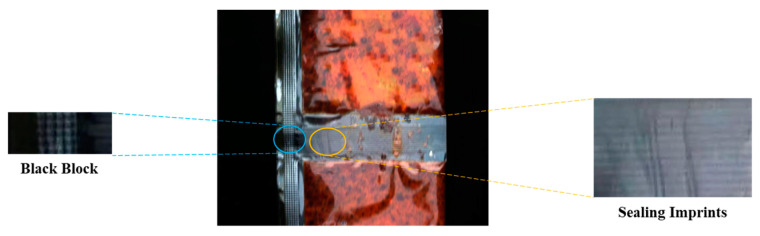
The black block and sealing imprints of a sauce packet.

**Table 1 sensors-24-03216-t001:** Detailed performance comparison of ablation experiment without attention mechanisms. Baseline: our proposed network without ULIE, the MFFM, and the attention mechanism; the backbone is EdgeNext. ULIE: uneven-light image enhancement method. MFFM: multi-scale feature fusion module.

Methods	UEIE	FPN [40]	AF-FPN [41]	MFFM	mIoU (%)	mPA (%)	F1-Score (%)	Params (M)	GFLOPS	FPS
Baseline					75.6	85.6	85.2	11.333	36.671	71.2
With MobileViTv2 [45]					73.9	84.3	83.9	11.278	36.879	63.2
With MobileNetv3 [46]					75.0	84.8	85.1	11.091	36.488	71.3
With MobileOneS4 [47]					75.3	83.6	84.9	22.951	40.312	36.2
With Regnetx [48]					75.1	86.2	84.2	12.471	36.674	70.8
With EfficientNetv2s [49]					76.3	85.9	86.8	29.872	40.113	42.8
+ULIE	✓				77.4	89.6	86.5	11.333	36.671	71.2
+R2RNet [21]					64.3	76.1	78.0	32.362	3007.287	0.6
+FPN		✓			77.3	85.2	86.3	18.906	1692.381	14.0
+AF-FPN			✓		75.3	83.6	84.8	18.908	1692.737	13.2
+MFFM				✓	78.7	87.9	**87.3**	11.564	66.736	48.8
+ULIE +FPN	✓	✓			78.5	86.7	87.2	11.906	1692.381	14.0
+ULIE +AF-FPN	✓		✓		75.9	84.3	85.3	11.908	1692.737	13.2
+ULIE +MFFM	✓			✓	79.2	89.1	87.7	11.564	66.736	48.8

**Table 2 sensors-24-03216-t002:** A detailed performance comparison of ablation experiments involving attention mechanisms. BUM represents the baseline with ULIE and the MFFM.

Methods	GAM [50]	SimAM [51]	SSAM [52]	mIoU (%)	mPA (%)	F1-Score (%)	Params (M)	GFLOPS	FPS
BUM				79.2	89.1	87.7	11.564	66.736	48.8
+GAM	✓			78.4	89.2	87.7	20.273	119.349	3.6
+SimAM		✓		77.1	86.6	86.2	11.564	66.736	48.8
+SSAM			✓	80.8	90.1	88.8	12.266	69.973	35.2

**Table 3 sensors-24-03216-t003:** Evaluation results of our ISLS and the SOTA CNN methods.

Methods	mIoU (%)	mPA (%)	F1-Score (%)	Params (M)
HRNet [13]	77.7	83.0	85.5	9.637
BiseNetv2 [14]	75.5	79.1	85.6	5.191
SegFormer [15]	76.5	80.8	85.1	3.715
PSPNet [55]	63.4	67.6	75.4	46.707
DeepLabv3 [56]	78.4	83.6	86.2	54.709
LIEPNet [23]	79.9	89.2	87.5	3.271
ISLS (Ours)	80.8	90.1	88.8	12.266

**Table 4 sensors-24-03216-t004:** Evaluation results of our ISLS and the traditional methods.

Methods	mIoU (%)	mPA (%)	F1-Score (%)
Template matching [61]	40.9	59.8	54.0
Canny edge segmentation [57]	32.5	44.5	43.6
Contour segmentation [58]	32.5	44.5	43.6
PCA segmentation [59]	36.6	58.6	50.4
iForest segmentation [60]	48.0	59.1	58.8
ISLS (ours)	80.8	90.1	88.8

**Table 5 sensors-24-03216-t005:** The ISIC dataset generalizability validation of our ISLS and the SOTA methods.

Methods	mIoU (%)	mPA (%)	F1-Score (%)	AUC (%)	Params (M)
HRNet [13]	78.4	86.6	88.2	91.6	9.637
BiseNetv2 [14]	76.9	84.1	86	89.4	5.191
SegFormer [15]	79.4	86	89.3	88.8	3.715
PSPNet [55]	76.4	83.3	86.9	89.1	46.707
DeepLabv3 [56]	75.8	85.1	85.8	85.0	54.709
LIEPNet [23]	79.7	85.6	89.0	91.8	3.271
AVGSC [62]	-	-	-	91.3	-
L-SVM [62]	-	-	-	92.6	-
NL-SVM [62]	-	-	-	90.4	-
Unext [66]	81.7		89.7		1.470
DoubleU-Net [67]	82.1	-	91.1	-	-
E-SegNet [68]	83.4	-	85.3	-	-
ISLS (Ours)	89.3	94.7	94.2	93.2	12.266

Note: The evaluation index information in Table 4 and Table 5 is provided in the related papers. Unfortunately, these related works do not have open-source code. We are unable to obtain the other evaluation indexes (indicated by ‘-’).

**Table 6 sensors-24-03216-t006:** The RSDDs dataset generalizability validation of our ISLS and the SOTA methods.

Methods	mIoU (%)	mPA (%)	F1-Score (%)	AUC (%)	Params (M)
HRNet [13]	73.0	80.0	81.5	75.9	9.637
BiseNetv2 [14]	71.2	73.7	74.3	63.4	5.191
SegFormer [15]	65.7	70.1	71.3	66.8	3.715
PSPNet [55]	70.0	74.6	75.2	65.3	46.707
DeepLabv3 [56]	72.1	81.6	75.7	71.2	54.709
LIEPNet [23]	74.4	82.5	82.8	78.9	3.271
MLC + PEME [63]	-	-	75.7	-	-
CFE [63]	-	-	85.1	-	-
NDD-Net [64]	-	-	-	88.2	-
CCEANN [65]	-	-	92.0	-	167.280
PFCNN [69]	-	-	82.9	-	5.000
ISLS (Ours)	75.8	85.0	85.2	88.4	12.266

Notably: The evaluation index information in Table 4 and Table 5 is provided in the related papers. Unfortunately, these related works do not have open-source code. We are unable to obtain the other evaluation indexes (indicated by ‘-’).

## Data Availability

The data that support the findings of this study are available from the corresponding author, [Ji, Liu], upon reasonable request. Our code is available at https://github.com/LSJ5106/SauceDetect (accessed on 7 March 2024).

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
