# Peer review of "ISLS: An Illumination-Aware Sauce-Packet Leakage Segmentation Method"

_sensors, 2024, doi:10.3390/s24103216_

Round 1

Reviewer 1 Report

Comments and Suggestions for Authors

The paper introduces the Illumination-aware Sauce-packet Leakage Segmentation (ISLS) method, addressing the challenge of imaging blurring due to uneven illumination in sauce-packet leakage detection on intelligent sensors, improving segmentation performance significantly.

Could you explain how the ISLS method might adapt to different or evolving packaging designs and materials that could affect the illumination and, thus, the segmentation performance?

How does the ISLS approach handle extreme cases of sauce-packet damage where the leakage is at the seams and across the packet surface?

What considerations were made in designing the Multi-scale Feature Fusion Module (MFFM) and Sequential Self-Attention Mechanism (SSAM) to ensure they remain computationally efficient while delivering high segmentation accuracy?

Could the authors detail the limitations or potential biases in the self-built dataset used for training and validation, and how might these impact the generalizability of the ISLS method to other similar industrial applications?

The experimental results are convincing. The performance proves the superiority of the proposed method.

References provided in the reference section are outdated, insufficient, and incomplete, must be updated. 

Overall the work is of good quality and the obtained results are interesting, but the above points should be addressed to improve the current version and to iron out the gaps in the study for further experimentation.

Author Response

We would like to thank the reviewer and editor for their careful reading of the manuscript and for their valuable comments.  The detailed modifications are displayed in the document. In the revised paper, the revised parts are marked with blue.   In the following explanation of revisions, the reviewer’s comments are provided under the heading of “Question”.

Reviewer 2 Report

Comments and Suggestions for Authors

To solve the sauce packaging leakage detection problem, this article designs a two-stage model that handles detection ROI and sauce leakage image segmentation, respectively. It also adds the so-called MFFM, SSAM enhancement methods, and the derivation of ULIE. This article seems to be in line with technically sound. But is it necessary to use a segmentation model for sauce leakage detection? I remain in doubt. Since the sauce leakage in this article is limited to the separation area between two sauce packages and the data and lighting noise presented, it can be processed only using a small image classification CNN or traditional image processing.  So, the research motivation and experimental design are unconvincing. However, if the model proposed in this article performs well in segmenting blurry images, it is strongly recommended that the author use the medical image segmentation data set for evaluation and then submit it to a journal in the field of medical image segmentation. This will be more appropriate and consistent with the excellent performance presented in this article.

other issues:

This article mentions using Yolo for ROI detection but has not seen relevant data set presentation and model training.

Please explain how MFFM and SSAM are embedded in the backbone's EdgeNext and provide an architecture diagram after adding them to the original EdgeNext architecture.

Please add and compare the evaluation data of EdgeNext in Tabel3.

Figure 9-10 shows an apparent difference between the Ground truth and the input image. Please add at least 4 samples of different patterns and add the outlines of the Ground truth and the input image.

section 2 Materials and methods, How many images are there with and without sauce leakage?

section 2 Materials and methods, please explain the original image size, and the image size of the input and output of the two detection stages respectively.

Finally, it is essential to avoid the "For AI's Sake" phenomenon and focus on addressing real needs and solving actual problems. This ensures that the application of AI technology is meaningful and effective.

Author Response

(The authors gave the same response as above.)

Reviewer 3 Report

Comments and Suggestions for Authors

  1. The main question addressed by the research:
    The paper proposes the two stage Illumination-aware Sauce-packet Leakage Segmentation (ISLS) method. First stage  the Uneven-Light Image Enhancement (ULIE) method suggested to alleviate the problems of blurred images under uneven illumination conditions. 
    Second stage: Multi-scale Feature Fusion Module (MFFM) for capturing multi-scale discriminative representation is suggested
  2. The suggestd topic seems to be original. However, due to the due to the careless use of literature review I can not conclude a lot of  or relevance of the topic in the field.
    As authors stay the topic it address to issues of overexposure or insufficient illumination, resulting in blurred images which can lead to reduction of the segmentation metrics. As for me some detailed example of the problem in application point of view should ge given here, as well as image examples. From the text problem seems to be not so scientifically impoartant.
  3. ALso the intro section does not allow to conclude about SOTA in the field. The specificity of the sauce-packet's leakage task does not shown convincingly enogth in the introduction. More over, The references 2,3,5 corresponds to the anomaly detection task. Than authors does not answering on the question "What does it add to the subject area compared with other published material?"
  4. The fisrt text of the section 2 should be named. Moreover fig.1 are not readbly enoght. Moreover in the text authors stay: "We propose Illumination-aware Sauce-packet Leakage Segmentation (ISLS) for the industrial production line, as illustrated in Figure 1."  However, Fig. 1 something like illustration of all possible variants of research authors had. The fig. 1 and its description should be clarified.
  5. In fig. 7 threshold should be written for x axis. In Fig.7 comment some explanation of threshold and value should be given. The same for fig.8 and their comaprision.
  6. some discussion on all experiment should be added.
  7. Also  as one of the contribution authors stay that  generate a new dataset on wich thay obtain metrics higher than there previous model, which was designed for another problem. In this sense achivement is doubtfull. I can suggest to rest only new dataset and contribution. Moreover if authors stay their dataset as contribution thay should provide link or refferance to prove that.
  8. The conclusions consistent with the evidence and arguments presented only in some sence. The numerical results and conclusion in the terms of application should be added to address the main question of the work.
  9. Some order in the references should also be done.
  10. The text in Fig.2 should be written as text not as blured image.
  11. Fig. 11 could be replaced with table.

Author Response

(The authors gave the same response as above.)

Round 2

Reviewer 3 Report

Comments and Suggestions for Authors

1 Instead of concise and specific answers with pointing on  corrected paper parts authors have decide to prepare one amlost new paper of 25 pages with extensive exhausted responses! Authors additionally should work on reprasentation of responces in the main paper text where it was not done.

2 More over, most of the information authors try to explain in the responces should also be included into the paper

3 The authors statement "We are the first one to study sauce-packet leakage detection under uneven illumination conditions" confirm my guess that the problem has not so much scientific interest.  If the authors object, they should write very detailed arguments in the paper. Morevoer, related task should be named in the text of paper. ALso the answers on "2) Regarding the detailed example of the problem in application point of view" and "3) Regarding to the text problem seems to be not so scientifically important" сan be taken as base for arguments.

4 The same as above for statement " to the best of our knowledge, we found that our ISLS method is the first research to detect the sauce-packet leakage." 

The tables 4 and 5 are some steps in the aforementioned direction.

5 The same about explanations for the question 5.
6 For so large experiment part the discussion section should be added as extra section.

7 if authors have open github with dataset than they may put it in to convinient data storage (like mendelay or other) with link in the paper.

8 why authors does not want to put Figure. Overall ISLS flow. into the paper?

Author Response

We appreciate your meticulous review and valuable feedback. 

Overall Paper Modify:

  1. Line 16-18: Define the task 'The Blurring Sauce-packet Leakage Segmentation Task (BSLST)' in the abstract.
  2. Line 42-59: Rewrite the introduction to include the scientific significance of the blurring sauce-packet leakage segmentation task.
  3. Line 93-96: Include a description stating, “To the best of our knowledge, we found that our ISLS method is the first research to detect the sauce-packet leakage under uneven illumination.”
  4. Line 112-115: Include an application description of sauce-packet leakage detection and meaning.
  5. Line 149-153: Include the overall ISLS flow figure and some related description.
  6. Line 387-389: Include the ULIE experiment figures’ description.
  7. Line 408-410: Modify the segmentation results with and without ULIE method figure.
  8. Line 559-584: Rearrange the text and table of the 'Generalization Performance Validation' experiment and create a new subsection 3.7.
  9. Line 585-620: Include a new subsection 3.8 discussion of all experiments.
